# Improving Self-supervised Pre-training *via* a Fully-Explored Masked Language Model

## Abstract

Masked Language Model (MLM) framework has been widely adopted for self-supervised language pre-training. In this paper, we argue that randomly sampled masks in MLM would lead to undesirably large gradient variance. Thus, we theoretically quantify the gradient variance via correlating the gradient covariance with the Hamming distance between two different masks (given a certain text sequence). To reduce the variance due to the sampling of masks, we propose a *fully-explored* masking strategy, where a text sequence is divided into a certain number of *non-overlapping* segments. Thereafter, the tokens within one segment are masked for training. We prove, from a theoretical perspective, that the gradients derived from this new masking schema have a smaller variance and can lead to more efficient self-supervised training. We conduct extensive experiments on both continual pre-training and general pre-training from scratch. Empirical results confirm that this new masking strategy can consistently outperform standard random masking. Detailed efficiency analysis and ablation studies further validate the advantages of our *fully-explored* masking strategy under the MLM framework.

## 1 Introduction

Large-scale pre-trained language models have attracted tremendous attention recently due to their impressive empirical performance on a wide variety of NLP tasks. These models typically abstract semantic information from massive unlabeled corpora in a self-supervised manner. Masked language model (MLM) has been widely utilized as the objective for pre-training language models. In the MLM setup, a certain percentage of words within the input sentence are masked out, and the model learns useful semantic information by predicting those missing tokens.

Previous work found that the specific masking strategy employed during pre-training plays a vital role in the effectiveness of the MLM framework (Liu et al., 2019; Joshi et al., 2019; Sun et al., 2019). Specifically, Sun et al. (2019) introduce entity-level and phrase-level masking strategies, which incorporate the prior knowledge within a sentence into its masking choice. Moreover, Joshi et al. (2019) propose to mask out random contiguous spans, instead of tokens, since they can serve as more challenging targets for the MLM objective.

Although effective, we identify an issue associated with the random sampling procedure of these masking strategies. Concretely, the difficulty of predicting each masked token varies and is highly dependent on the choice of the masking tokens. For example, predicting stop words such as "*the*" or "*a*" tends to be easier relative to nouns or rare words. As a result, with the same input sentence, randomly sampling certain input tokens/spans, as a typical masking recipe, will result in undesirable large variance while estimating the gradients. It has been widely demonstrated that large gradient variance typically hurts the training efficiency with stochastic gradient optimization algorithms (Zhang & Xiao, 2019; Xiao & Zhang, 2014; Johnson & Zhang, 2013). Therefore, we advocate that obtaining gradients with *a smaller variance* has the potential to enable more sample-efficient learning and thus accelerate the self-supervised learning stage.

In this paper, we start by introducing a theoretical framework to quantify the variance while estimating the training gradients. The basic idea is to decompose the total gradient variance into two terms, where the first term is induced by the data sampling process and the second one relates to the sampling procedure of masked tokens. Theoretical analysis on the second variance term demonstrates that it can be minimized by reducing the gradient covariance between two masked sequences.

Furthermore, we conduct empirical investigation on the correlation between the gradient's covariance while utilizing two masked sequences for training and the Hamming distance between these sequences. We observed that that the gradients' covariance tends to decrease monotonically *w.r.t* the sequences' Hamming distance.

Inspired by the observations above, we propose a *fully-explored* masking strategy, which maximizes the Hamming distance between any of two sampled masks on a fixed text sequence. First, a text sequence is randomly divided into multiple *non-overlapping* segments, where each token (*e.g.* subword, word or span) belongs to one of them. While the model processes this input, several different training samples are constructed by masking out one of these segments (and leaving the others as the contexts). In this manner, the gradient *w.r.t.* this input sequence can be calculated by averaging the gradients across multiple training samples (produced by the same input sequence). We further verify, under our theoretical framework, that the gradients obtained with such a scheme tend to have smaller variance, and thus can improve the efficiency of the pre-training process.

We evaluate the proposed masking strategies on both continued pre-training (Gururangan et al., 2020) and from-scratch pre-training scenarios. Specifically, Computer Science (CS) and News domain corpus (Gururangan et al., 2020) are leveraged to continually pre-train RoBERTa models, which are then evaluated by fine-tuning on downstream tasks of the corresponding domain. It is demonstrated that the proposed fully-explored masking strategies lead to pre-trained models with stronger generalization ability. Even with only a subset of the pre-training corpus utilized in (Gururangan et al., 2020), our model consistently outperforms reported baselines across four natural language understanding tasks considered. Besides, we also show the effectiveness of our method on the pre-training of language models *from scratch*. Moreover, the comparison between fully-explored and standard masking strategies in terms of their impacts on the model learning efficiency further validates the advantages of the proposed method. Extensive ablation studies are further conducted to explore the robustness of the proposed masking scheme.

## 2 RELATED WORK

**Self-supervised Language Pre-training**   Self-supervised learning has been demonstrated as a powerful paradigm for natural language pre-training in recent years. Significant research efforts have been devoted to improve different aspects of the pre-training recipe, including training objective (Lewis et al., 2019; Clark et al., 2019; Bao et al., 2020; Liu et al., 2019), architecture design (Yang et al., 2019; He et al., 2020), the incorporation of external knowledge (Sun et al., 2019; Zhang et al., 2019), *etc*. The idea of self-supervised learning has also been extended to generation tasks and achieves great results (Song et al., 2019; Dong et al., 2019). Although impressive empirical performance has been shown, relatively little attention has been paid to the efficiency of the pre-training stage. ELECTRA (Clark et al., 2019) introduced a discriminative objective that is defined over all input tokens. Besides, it has been showed that incorporating language structures (Wang et al., 2019) or external knowledge (Sun et al., 2019; Zhang et al., 2019) into pre-training could also help the language models to better abstract useful information from unlabeled samples.

In this work, we approach the training efficiency issue from a different perspective, and argue that the masking strategies, as an essential component within the MLM framework, plays a vital role especially in efficient pre-training. Notably, our *fully-explored* masking strategies can be easily combined with different model architectures for MLM training. Moreover, the proposed approach can be flexibly integrated with various tokenization choices, such as subword, word or span (Joshi et al., 2019). A concurrent work Chen et al. (2020) also shared similar motivation as this work, although they have a different solution and their method requires additional computation to generate the masks, and yet is outperformed by the proposed *fully-explored* masking (see Table 2).

**Domain-specific Continual Pre-training**   The models mentioned above typically abstract semantic information from massive, heterogeneous corpora. Consequently, these models are not tailored to any specific domains, which tends to be suboptimal if there is a domain of interest beforehand. Gururangan et al. (2020) showed that continual pre-training (on top of general-purpose LMs) with in-domain unlabeled data could bring further gains to downstream tasks (of that particular domain). One challenge inherent in continual pre-training is that in-domain data are usually much more limited, compared to domain-invariant corpora. As a result, how to efficiently digest information from unlabeled corpus is especially critical while adapting large pre-trained language models to specific

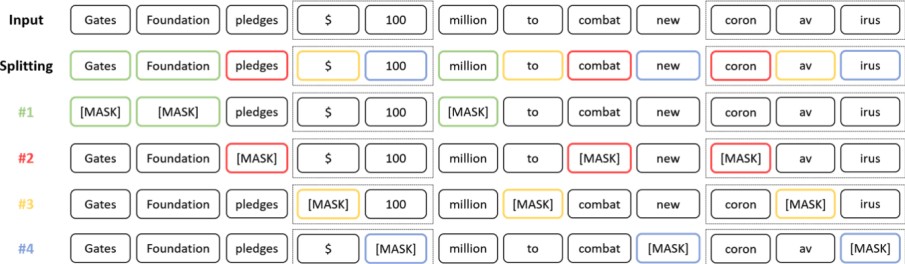

Figure 1: Illustration of the proposed fully-explored masking strategy with a specific example. In this case, the input sequence has been divided into 4 exclusive segments, where different colors indicate which segment a certain token belongs to.

domains. To this end, we specifically consider the continual pre-training scenario to evaluate the effectiveness of our approach.

## 3 PROPOSED APPROACH

In this section, we first review the MLM framework that is widely employed for natural language pre-training. Motivated by the gradient variance analysis of MLM in section 3.2, we present the *fully-explored* masking strategy, which serves as a simple yet effective solution to reduce the gradient variance during training. Connections between our method and variance reduction theory are further drawn, which provides a theoretical foundation for the effectiveness of the proposed strategy. Finally, some specific implementation details are discussed.

### 3.1 BACKGROUND: THE MLM FRAMEWORK

Let $V$ denote the token vocabulary and $\mathbf{x} = (x_1, \ldots, x_n)$ denote a sentence of $n$ tokens, where $x_i \in \mathcal{V}$ for $i = 1, \ldots, n$. Let $\mathbf{m} = (m_1, \ldots, m_n)$ denote a binary vector of length $n$, where $m_i \in \{0, 1\}$, representing the mask over a sentence. Specifically, $m_i = 1$ means the token $x_i$ is masked and $m_i = 0$ if $x_i$ is not masked. We use $\mathbf{m} \circ \mathbf{x}$ to denote a masked sentence, that is,

$$(\mathbf{m} \circ \mathbf{x})_i = \begin{cases} \texttt{[MASK]} & \text{if } m_i = 1, \\ x_i & \text{if } m_i = 0. \end{cases}$$

In addition, let $\overline{\mathbf{m}}$ be the complement of $\mathbf{m}$; in other words, $\overline{m}_i = 0$ if $m_i = 1$ and $\overline{m}_i = 1$ if $m_i = 0$. Naturally, $\overline{\mathbf{m}} \circ \mathbf{x}$ denotes a sentence with the complement mask $\overline{\mathbf{m}}$.

For a typical language model with parameters $\theta$, its loss function over a sentence $\mathbf{x} \in \mathcal{V}^n$ and a mask $\mathbf{m} \in \{0, 1\}^n$ as

$$\ell(\theta; \mathbf{x}, \mathbf{m}) = -\log P(\overline{\mathbf{m}} \circ \mathbf{x} \,|\, \theta, \mathbf{m} \circ \mathbf{x}) = -\sum_{i\,:\,m_i=1} \log P(x_i \,|\, \theta, \mathbf{m} \circ \mathbf{x}), \tag{1}$$

where $P(x_i \,|\, \theta, \mathbf{m} \circ \mathbf{x})$ is the probability of the model correctly predicting $x_i$ given the masked sentence $\mathbf{m} \circ \mathbf{x}$. If $m_i = 0$, it always has $P(x_i \,|\, \theta, \mathbf{m} \circ \mathbf{x}) = 1$ as the ground-truth $x_i$ is not masked.

We will focus on masks with a fixed length. Let $\tau$ be an integer satisfying $0 \leq \tau \leq n$. The set of possible masks of length $\tau$ is defined as $\mathcal{M}(\tau)$,

$$\mathcal{M}(\tau) = \big\{ \mathbf{m} \in \{0, 1\}^n \mid \textstyle\sum_{i=1}^n m_i = \tau \big\},$$

which has a cardinality $|\mathcal{M}(\tau)| = \binom{n}{\tau} = \frac{n!}{\tau!(n-\tau)!}$. Therefore, the average loss function over a sentence $\mathbf{x}$ with masks of length $\tau$ is,

$$L(\theta; \mathbf{x}) = \mathbb{E}_{\mathbf{m} \sim \text{Unif}(\mathcal{M}(\tau))} \ell(\theta; \mathbf{x}, \mathbf{m}) = \frac{1}{\binom{n}{\tau}} \sum_{\mathbf{m} \in \mathcal{M}(\tau)} \ell(\theta; \mathbf{x}, \mathbf{m}). \tag{2}$$

Let's consider $P_{\mathcal{D}}$ as the probability distribution of sentence in a corpus $\mathcal{D} \subset \mathcal{V}^n$. The overall loss function for training the masked language model over corpus $\mathcal{D}$ is

$$L(\theta) \triangleq \mathbb{E}_{\mathbf{x} \sim P_{\mathcal{D}}} L(\theta; \mathbf{x}) = \mathbb{E}_{\mathbf{x} \sim P_{\mathcal{D}}} \mathbb{E}_{\mathbf{m} \sim \text{Unif}(\mathcal{M}(\tau))} \ell(\theta; \mathbf{x}, \mathbf{m}). \tag{3}$$

During each step of the training process, it randomly samples a mini-batch of sentences $\mathcal{S}_t \subset \mathcal{D}$. For each $\mathbf{x} \in \mathcal{S}_t$, we randomly pick a subset of masks $\mathcal{K}_t(\mathbf{x}) \subset \mathcal{M}(\tau)$, independently across different $x$. Thus, the mini-batch stochastic gradient is

$$g_t(\theta) = \frac{1}{S} \sum_{\mathbf{x} \in \mathcal{S}_t} \frac{1}{K} \sum_{\mathbf{m} \in \mathcal{K}_t(\mathbf{x})} \nabla_\theta \ell(\theta; \mathbf{x}, \mathbf{m}). \tag{4}$$

where $|\mathcal{S}_t| = S$ and $|\mathcal{K}_t(\mathbf{x})| = K$ for all $t$. Clearly we have $\mathbb{E}[g_t(\theta)] = \nabla L(\theta)$. In the following sections, it first gives the variance of $g_t(\theta)$ which is an important factor to influence model training efficiency (Xiao & Zhang, 2014; Zhang & Xiao, 2019), and then it presents the proposed *fully-explored* masking strategy to help reduce the gradient variance of the masked language model.

## 3.2 ANALYSIS: GRADIENT VARIANCE OF MLM

According to the law of total variance (Weiss, 2005), the variance of the mini-batch stochastic gradient $\mathrm{Var}_{\mathcal{S}_t, \mathcal{K}_t}(g_t)$ can be decomposed as follows,

$$\mathrm{Var}_{\mathcal{S}_t, \mathcal{K}_t}(g_t) = \mathbb{E}_{\mathcal{S}_t}\left[\mathrm{Var}_{\mathcal{K}_t}(g_t) \mid \mathcal{S}_t\right] + \mathrm{Var}_{\mathcal{S}_t}\left(\mathbb{E}_{\mathcal{K}_t}\left[g_t \mid \mathcal{S}_t\right]\right), \tag{5}$$

where for simplicity $g_t$ indicates $g_t(\theta)$ as in eqn. 4, the first term captures the variance due to the sampling of masks, and the second term is the variance due to the sampling of mini-batch sentences.

In this work, we focus on the analysis of the first term in eqn. 5: the variance due to the sampling of masks. Denote $g(\mathbf{m}) = \nabla_\theta \ell(\theta; \mathbf{x}, \mathbf{m})$ for any fixed sentence $\mathbf{x}$. Consider a subset of random masks $(\mathbf{m}_1, \ldots, \mathbf{m}_K)$ and the $K$-masks gradient is defined as the average of them:

$$g(\mathbf{m}_1, \ldots, \mathbf{m}_K) = \frac{1}{K} \sum_{k=1}^{K} g(\mathbf{m}_k). \tag{6}$$

**Theorem 1.** *The Variance of $K$-masks gradient:* $\mathrm{Var}\big(g(\mathbf{m}_1, \ldots, \mathbf{m}_K)\big)$ *is*

$$\frac{1}{K}\mathrm{Var}\big(g(\mathbf{m}_1)\big) + \left(1 - \frac{1}{K}\right)\mathrm{Cov}\big(g(\mathbf{m}_1), g(\mathbf{m}_2)\big). \tag{7}$$

where,

$$\mathrm{Cov}\big(g(\mathbf{m}_1), g(\mathbf{m}_2)\big) = \mathbb{E}\left[\big(g(\mathbf{m}_1) - \bar{g}\big)^T \big(g(\mathbf{m}_2) - \bar{g}\big)\right], \tag{8}$$

and

$$\bar{g} = \mathbb{E}_{\mathbf{m} \sim \mathrm{Unif}(\mathcal{M}(\tau))} g(\mathbf{m}) = \frac{1}{\binom{n}{\tau}} \sum_{\mathbf{m} \in \mathcal{M}(\tau)} g(\mathbf{m}). \tag{9}$$

The detailed proof of Theorem 1 is given in Appendix A.1. In the theorem 1, it indicates that the variance of $K$-masks gradient can be reduced by decreasing the gradient covariance between different masks.

## 3.3 VARIANCE REDUCTION: FULLY-EXPLORED MASKING

Intuitively, if we consider the two random masks $\mathbf{m}_1$ and $\mathbf{m}_2$ are totally overlapped, the gradient covariance between them should be maximized. It motivates us to consider the correlation between gradient covariance and Hamming distance between these two masks. Thus, we have the following assumption:

**Assumption 1.** *The covariance* $\mathrm{Cov}\big(g(\mathbf{m}_1), g(\mathbf{m}_2)\big)$ *is monotone decreasing in term of the Hamming distance between* $\mathbf{m}_1$ *and* $\mathbf{m}_2$.

To verify the Assumption 1, we sample a small set of CS domain sentences from S2ORC dataset (Gururangan et al., 2020) as the fixed mini-batch for our analysis, then calculate gradient covariance $\mathrm{Cov}\big(g(\mathbf{m}_1), g(\mathbf{m}_2)\big)$ of mask pairs $(\mathbf{m}_1, \mathbf{m}_2)$ with different Hamming distances $H(\mathbf{m}_1, \mathbf{m}_2)$ using this mini-batch. In Figure 2, the center of gradient covariance distribution is shifting to left (lower value) as Hamming distance increases. In Figure 3, we also observe that the average gradient covariance is decreasing in term od Hamming distance. As shown in Figure 2, 3, Assumption 1 holds for both RoBERTa-base model (Liu et al., 2019) and RoBERTa-base model after continually pre-trained on CS domain corpus.

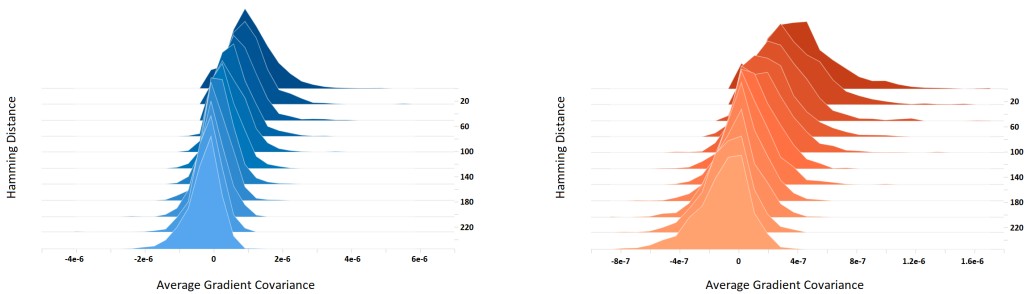

Figure 2: The distributions of gradient covariance $\text{Cov}\big(g(\mathbf{m}_1), g(\mathbf{m}_2)\big)$ for different Hamming distances $H(\mathbf{m}_1, \mathbf{m}_2)$ based on a small CS domain corpus. Left: gradient covariance distribution of selected parameters in RoBERTa-base model; Right: gradient covariance distribution of selected parameters in RoBERTa-base model after continually pre-trained on CS domain corpus.

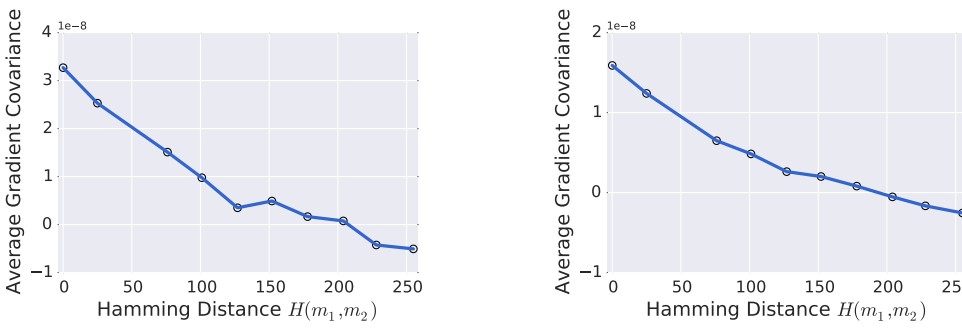

Figure 3: Empirical analysis of the correlation between gradient covariance $\text{Cov}\big(g(\mathbf{m}_1), g(\mathbf{m}_2)\big)$ and Hamming distance $H(\mathbf{m}_1, \mathbf{m}_2)$ based on a small CS domain corpus. For a sequence of length 512, two masks $\mathbf{m}_1$, $\mathbf{m}_2$ are randomly sampled with 128 masked tokens, their Hamming distance satisfying $0 \leq H(\mathbf{m}_1, \mathbf{m}_2) \leq 256$. Left: gradient covariance calculated based on RoBERTa-base model; Right: gradient covariance calculated based on RoBERTa-base model after continually pre-trained on CS domain corpus.

We propose the *fully-explored* masking strategy which restricts masks sampled from $\mathcal{M}(\tau)$ to be non-overlapping, denoted as $\mathcal{M}_{\text{FE}}(\tau)$ for simplicity:

$$\mathbf{m}_1, \ldots, \mathbf{m}_K \sim \mathcal{M}_{\text{FE}}(\tau), \forall_{i \neq j} H(\mathbf{m}_i, \mathbf{m}_j) = 2\tau \tag{10}$$

With the *fully-explored* masking strategy, it can be easily approved that expectation of gradient over $\mathcal{M}_{\text{FE}}(\tau)$ is an unbiased estimation of the expectation of gradient over $\mathcal{M}(\tau)$ as in the Lemma 2. In the Lemma 3, it states that the Theorem 1 is still hold for *fully-explored* masking strategy, which indicates that the variance of K-masks gradient can be reduced by restricting the masks sampling from $\mathcal{M}_{\text{FE}}(\tau)$.

**Lemma 2.** *The expectation of gradient over $\mathcal{M}_{FE}(\tau)$ equals to the expectation of gradient over $\mathcal{M}(\tau)$.*

*Proof.* The joint distributions of $(\mathbf{m}_1, \ldots, \mathbf{m}_K)$ sampling from $\mathcal{M}_{\text{FE}}(\tau)$ is different from the i.i.d. case due to the non-overlapping restriction. However, the marginal distributions of $\mathbf{m}_k$ are still the same uniform distribution over $\mathcal{M}(\tau)$. Therefore, we still have $\mathbb{E}\big[g(\mathbf{m}_k)\big] = \bar{g}, \forall_{k=1,\ldots,K}$ and as a consequence $\mathbb{E}\big[g(\mathbf{m}_1, \ldots, \mathbf{m}_K)\big] = \bar{g}$. $\qquad\square$

**Lemma 3.** *The derivation of K-masks gradient variance in Eqn.7 holds for both $\mathcal{M}_{FE}(\tau)$ and $\mathcal{M}(\tau)$.*

The detailed proof of Lemma 3 can be seen in Appendix A.2.

### 3.4 IMPLEMENTATION DETAILS

The details of *fully-explored* masking algorithm is illustrated in Algorithm 1. In practice, a text sequence $\mathcal{S}_i$ is tokenized into subword pieces (Devlin et al., 2018) with the maximum sequence

length $n$ set as $512$ in the experiments. To understand the performance of *fully-explored* masking strategy at different granularity, the text sequence $\mathcal{S}_i$ is masked at both subword level (Devlin et al., 2018; Liu et al., 2019) and span level (Joshi et al., 2019; Wang et al., 2019). The details about other hyperparameters, i.e., masking-ratio and number of splits $K$ will be discussed in experiment section.

---

**Algorithm 1:** Fully-explored Masking Language Model

---

**Input:** Language corpus $\mathcal{D} = \{\mathcal{S}_1, ..., \mathcal{S}_T\}$, $|\mathcal{S}_i| = n$; Masking ratio $\frac{\tau}{n}$; Number of sampling
      masks $K$, where $K * \frac{\tau}{n} \leq 1$; Initial model parameters $\theta_0$;
**Output:** model parameters $\theta^*$
**foreach** $\mathcal{S}_i \in \mathcal{S}$ **do**
    |  Sample K split masking vectors $(\boldsymbol{m}_1, ..., \boldsymbol{m}_K)$ from $\mathcal{M}_{\text{FE}}(\tau)$ as in Eqn.10.
    |  Calculate the gradient $g(\boldsymbol{m}_1, ..., \boldsymbol{m}_K)$ as in Eqn. 6.
    |  Update model parameters $\theta_{i+1} = \text{Optimizer}(\theta_i, g(\boldsymbol{m}_1, ..., \boldsymbol{m}_K))$
**end**
 return $\theta^* = \theta_T$

---

## 4 EXPERIMENTS

In this section, we evaluate the proposed *fully-explored* masking strategy for natural language pre-training in two distinct settings: *i*) continual pre-training, where a given pre-trained model is further adapted leveraging domain-specific unlabeled corpus; *ii*) pre-training from scratch, where large-scale corpus such as Wikipedia and BookCorpus are employed to pre-train a model from the beginning. We also compare the training efficiency of FE-MLM and MLM frameworks to validate our theoretical findings. Ablation studies and analysis are further conducted regarding the proposed approach.

### 4.1 EXPERIMENTAL SETTINGS

For the continual pre-training scenario, we consider unlabeled corpus from two different domains, *i.e.*, computer science (CS) papers and news text from RealNews, introduced by Gururangan et al. (2020). As to the downstream tasks, ACL-ARC citation intent Jurgens et al. (2018) and SciERC relation classification Luan et al. (2018) are utilized for the CS domain. While for the News domain, HyperPartisan news detection Kiesel et al. (2019) and AGNews Zhang et al. (2015) are employed to facilitate the comparison with Gururangan et al. (2020).

Following (Gururangan et al., 2020) for a fair comparison, RoBERTa Liu et al. (2019) is leveraged as the initial model for continual pre-training, where the same training objective is optimized on the domain-specific corpus. We choose a batch size of $48$, and the model is trained using Adam Kingma & Ba (2014), with a learning rate of $1 \times 10^{-4}$. It is worth noting that we observe, in our initial experiments, that downsampling only 72k documents from the total of 2.22M used by Gururangan et al. (2020) can result in similar performance on downstream tasks. This happens in the News domain as well, where we randomly sample 623k documents out of 11.90M. The model is continually pre-trained for around 40k and 20k steps on the CS and News domain, respectively. One important hyperparameter under the FE-MLM framework is the number of split the input sequence is divided into, where we use $4$ as the default setting. The sensitivity of the proposed algorithm *w.r.t* this hyperparameter is further investigated (see Figure 4).

For the general pre-training experiments, we employ BERT as the baseline model. Wikiepdia and BookCorpus (Zhu et al., 2015) are used as the pre-training corpus, with a total size of 16G. We adopt the same tokenization (*i.e.*, WordPiece embeddings) as BERT, which consists of 30,522 tokens in the vocabulary. The model is optimized using Adam with the learning rate set as $1 \times 10^{-4}$. A batch size of 256 is employed, and we train the model for 1M step. The resulting model is evaluated on the GLUE benchmark (Wang et al., 2018), which comprises 9 natural language understanding (NLU) tasks such as textual entailment (MNLI, RTE), question-answer entailment (QNLI), question paraphrase (QQP), paraphrase (MRPC), sentimnt analysis (SST-2), linguistic acceptability (CoLA) and textual similarity (STS-B). The HuggingFace codebase[1] is used in our implementation for both settings.

---

[1]https://github.com/huggingface/transformers

| Model | ACL-ARC | SciERC | HyperPartisan | AGNews |
|---|---|---|---|---|
| RoBERTa (Gururangan et al., 2020) | $63.0 \pm 5.8$ | $77.3 \pm 1.9$ | $86.6 \pm 0.9$ | $93.9 \pm 0.2$ |
| DAPT (Gururangan et al., 2020) | $75.4 \pm 2.5$ | $80.8 \pm 1.5$ | $88.2 \pm 5.9$ | $93.9 \pm 0.2$ |
| MLM + Subword (our implementation) | $75.34 \pm 2.54$ | $81.51 \pm 1.05$ | $91.00 \pm 2.66$ | $94.05 \pm 0.16$ |
| FE-MLM + Subword | $76.24 \pm 1.86$ | **82.40**$\pm 0.86$ | $92.35 \pm 3.49$ | $94.02 \pm 0.09$ |
| MLM + Span (our implementation) | $76.63 \pm 1.65$ | $81.33 \pm 1.16$ | $91.72 \pm 3.26$ | $93.94 \pm 0.05$ |
| FE-MLM + Span | **78.06**$\pm 2.31$ | $81.99 \pm 0.79$ | **93.22**$\pm 3.31$ | **94.13**$\pm 0.04$ |

Table 1: The empirical results on continual pre-training setting, where RoBERTa and DAPT (RoBERTa continually pre-trained with the standard MLM objective) is leveraged as our baseline to facilitate comparison with (Gururangan et al., 2020). Specifically, ACL-ARC and SciERC are evaluated with the continually pre-trained model with CS domain corpus, while HyperPartisann and AGNews are based upon models trained with News domain corpus.

| Model | MNLI-m/mm | SST-2 | QNLI | QQP | RTE | MRPC | CoLA | STS-B |
|---|---|---|---|---|---|---|---|---|
| BERT (MLM) | 84.37/**84.85** | 92.78 | **91.01** | 91.09 | 63.54 | 87.01 | 59.65 | 87.89 |
| BERT (FE-MLM) | **85.09**/84.63 | **93.23** | **91.01** | **91.16** | **68.59** | **87.99** | **61.32** | **89.51** |

Table 2: The results on the dev sets of GLUE benchmarks, where MLM and FE-MLM are compared with the BERT-base model as the testbed.

## 4.2 EXPERIMENTAL RESULTS

**Continual Pre-training Evaluation** We applied our *fully-explored* MLM framework to both subword and span masking scenarios. The results for the RoEBRTa model continually pre-trained on the CS and News domains are presented in Table 1. It can be observed that the continual pre-training stage can benefit the downstream tasks on both domains(compared with fine-tuning the RoBERTa model directly). Besides, the baseline numbers based on our implementation is on par with or even better than those reported in Gururangan et al. (2020), even though we downsample the original unlabeled corpus (as described in the previous section).

More importantly, in the subword masking case, our FE-MLM framework consistently exhibits better empirical results on the downstream tasks. Note that to ensure fair comparison, the same computation is taken for both MLM and FE-LMLM training. This indicates that the models pre-trained using the FE-MLM approach have been endowed with stronger generalization ability, relative to standard MLM training. Similar trend is also observed in the span masking experiments, demonstrating that the proposed method can be naturally and flexibly integrated with different masking schemes. Besides, we found that subword masking tends to work better than span masking in the CS domain, whereas the opposite is true as to the News domain. This may be attributed to the different nature of the unlabeled corpus from two domains.

**General Pre-training Evaluation** We also evaluate the FE-MLM framework on the pre-training experiments with general-purpose unlabeled corpus. Specifically, we follow the same setting as BERT, except that the proposed *fully-explored* masking strategy is applied (the same amount of computation is used for the baseline and our method). The corresponding results are shown in Table 2. It can be found that the FE-MLM approach, while fine-tuned on the GLUE benchmark, exhibits better results on 7 out of 9 NLU datasets than the MLM baseline. This demonstrates the wide applicability of the proposed FE-MLM framework across different pre-training settings.

| Model | Avg. Score |
|---|---|
| ELMo, (Peters et al., 2018) | 71.2 |
| GPT, (Radford et al., 2018) | 78.8 |
| BERT-base, (Devlin et al., 2018) | 82.2 |
| BERT-base (ReEval) | 82.5 |
| MAP-Net, (Chen et al., 2020) | 82.1 |
| BERT-base (FE-MLM) | **83.6** |

Table 3: The comparison between the FE-MLM model with several baseline methods, based on the averaged score (on the dev set) across different tasks from the GLUE benchmark.

We further compare the averaged score over 9 GLUE datasets with other methods, and the numbers are summarized in Table 3. It is worth noting that the BERT-based (ReEval) baseline is obtained by fine-tuning the BERT model released by Devlin et al. (2018) on each GLUE datasets, with the results on the dev sets averaged. Another BERT-base number is reported by Clark et al. (2019), which is pretty similar our re-evaluation one. MAsk proposal network (MAP-Net) is proposed by Chen et al.

(2020), which shares the same motivation of reducing the gradient variance during the masking stage. However, we approach the problem with a distinct strategy based upon extensive theoretical analysis. We found that BERT-base model improved with the FE-MLM training significantly outperform BERT-base model and Mask Proposal Network, further demonstrating the effectiveness of proposed approach.

## 4.3 ABLATION STUDIES AND ANALYSIS

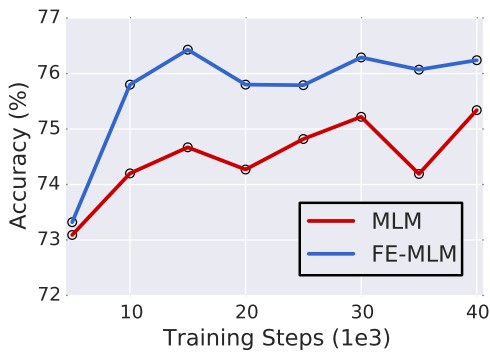 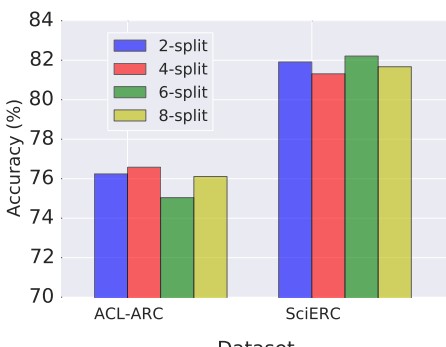

Figure 4: Left: the efficiency comparison between the standard MLM and *fully-explored* MLM approach. Specifically, the RoBERTa-base model and continual pre-training setting (on the CS domain) are employed. The corresponding models are evaluated on the ACL-ARC dataset (at different training steps). Right: the effect of split number (under the FE-MLM framework) on the generalization ability of pre-trained models, evaluated on two datasets in the CS domain.

**Training Efficiency** Although previous results has demonstrated that our model at the end of pre-training process exhibits stronger generalization ability, it is still unclear how the proposed FE-MLM framework influence the training efficiency during training. In this regard, we examine the intermediate models obtained with both MLM and FE-MLM training by fine-tuning and evaluating them on the ACL-ARC dataset. Specifically, the RoBERTa-base setting is used here, which is continually pre-trained on the unlabeled corpus from the CS domain. As shown on the left side of Figure 4, FE-MLM beats MLM at different steps of pre-training. More importantly, the performance of the FE-MLM model improves much faster at the early stage (*i.e.*, less than around 15,000 steps), indicating that the model is able to extract useful semantic information (from unlabeled corpus) more efficiently with the proposed masking strategy. This observation further highlights the advantage and importance of reducing gradient variance under the MLM framework.

**The Effect of Masking Split Number** The number of masking split the input sentence is divided into is a vital hyperparameter for the FE-MLM approach. Therefore, we investigate its impact on the performance of resulting models. Concretely, the setting of continual pre-training on the CS domain is employed, where the RoBERTa model is pre-trained with the FE-MLM objective. Different split number is explored, including $2, 4, 6, 8$, and $12.5\%$ of all the tokens are masked within each split. The results are visualized on the right side of Figure 4. We found that the downstream task performance (on both ACL-ARC and SciERC datasets) is fairly stable *w.r.t.* different split numbers. This may relate to our non-overlapping sampling strategy, which helps the model to explore various position in the sentence as efficiently as possible, so that the model exhibits strong performance even with only two splits.

## 5 CONCLUSION

In this paper, we identified that under the MLM framework, the procedure of randomly sampling masked tokens will give rise to undesirably large variance while estimating the training gradients. Therefore, we introduced a theoretical framework to quantify the gradient variance, where the connection between gradient covariance and the Hamming distance between two different masked sequences are drawn. Motivated by these observations, we proposed a *fully-explored* masking strategy, where a text sequence is divided into multiple non-overlapping segments. During training, all tokens in one segment are masked out, and the model is asked to predict them with the other segments as the context. It was demonstrated theoretically that the gradients obtained with such a novel masking strategy have a smaller variance, thus enabling more efficient pre-training. Extensive experiments on both continual pre-training and general pre-training from scratch showed that the proposed masking strategy consistently outperforms standard random masking.

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

## A  APPENDIX

### A.1  PROOF OF THEOREM 1

*Proof.*

$$
\begin{aligned}
\mathrm{Var}\big(g(\mathbf{m}_1,\ldots,\mathbf{m}_K)\big) &= \mathbb{E}\left[\left\|g(\mathbf{m}_1,\ldots,\mathbf{m}_K)-\bar{g}\right\|^2\right] \\
&= \mathbb{E}\left[\left\|\tfrac{1}{K}\sum_{k=1}^{K}g(\mathbf{m}_k)-\bar{g}\right\|^2\right] \\
&= \frac{1}{K^2}\mathbb{E}\left[\left\|\sum_{k=1}^{K}\big(g(\mathbf{m}_k)-\bar{g}\big)\right\|^2\right] \\
&= \frac{1}{K^2}\mathbb{E}\left[\sum_{k=1}^{K}\|g(\mathbf{m}_k)-\bar{g}\|^2 + \sum_{k\neq l}\big(g(\mathbf{m}_k)-\bar{g}\big)^T\big(g(\mathbf{m}_l)-\bar{g}\big)\right] \\
&= \frac{1}{K^2}\left(\sum_{k=1}^{K}\mathrm{Var}\big(g(\mathbf{m}_k)\big) + \sum_{k\neq l}^{K}\mathrm{Cov}\big(g(\mathbf{m}_k),g(\mathbf{m}_l)\big)\right)
\end{aligned}
\tag{11}
$$

where for each pair $k \neq l$,

$$
\mathrm{Cov}\big(g(\mathbf{m}_k),g(\mathbf{m}_l)\big) = \mathbb{E}\left[\big(g(\mathbf{m}_k)-\bar{g}\big)^T\big(g(\mathbf{m}_l)-\bar{g}\big)\right],
$$

Since $\mathbf{m}_1,\ldots,\mathbf{m}_K$ are i.i.d. samples from the uniform distribution over $\mathcal{M}(\tau)$, we have

$$
\mathrm{Var}\big(g(\mathbf{m}_1)\big) = \cdots = \mathrm{Var}\big(g(\mathbf{m}_K)\big)
\tag{12}
$$

$$
\mathrm{Cov}\big(g(\mathbf{m}_k),g(\mathbf{m}_l)\big) = \mathrm{Cov}\big(g(\mathbf{m}_1),g(\mathbf{m}_2)\big), \forall\, k \neq l.
\tag{13}
$$

Therefore we have the following variance decomposition:

$$
\mathrm{Var}\big(g(\mathbf{m}_1,\ldots,\mathbf{m}_K)\big) = \frac{1}{K}\mathrm{Var}\big(g(\mathbf{m}_1)\big) + \left(1-\frac{1}{K}\right)\mathrm{Cov}\big(g(\mathbf{m}_1),g(\mathbf{m}_2)\big).
\tag{14}
$$

□

## A.2  PROOF OF LEMMA 3

*Proof.* The joint distribution of the pairs $(\mathbf{m}_k, \mathbf{m}_l)$ sampling from $\mathcal{M}_{\mathrm{FE}}(\tau)$ are different from the i.i.d. case, it can be shown (by symmetry) that the identity equation 13 also holds. Considering the fact that the derivation in equation 11 holds for any sampling strategy, we conclude that the variance decomposition in equation 14 still holds. □

