# OpenReview forum: "Improving Self-supervised Pre-training via a Fully-Explored Masked Language Model"
_ICLR.cc/2021/Conference — Reject_

### Official Review · AnonReviewer4 · 2020-10-24
**Interesting, but experiments and notations can be strengthened.**

**Rating:** 5
**Confidence:** 3

**Review:**

To reduce the variance due to the sampling of masks, the authors propose a fully-explored masking strategy, where a text sequence is divided into a certain number of non-overlapping segments. And they show this technique improves accuracy in downstream tasks.

This idea is novel and interesting to me, and the derivation and experiment results look encouraging. However, I feel that experiments can be strengthened, and notations can be improved. Below are my major concerns:

If the major motivation is to reduce gradient variance, can we just use larger mini-batch size? In the experiments, it is not reported that the learning rate or the mini-batch size is well tuned for the baseline. I'd like some confirmation that larger batch size won't get much improvement for the baseline model. For example, in the continual learning, you said "choose a batch size of 48", that seems to be small to me?

In algorithm 1, in each iteration, only data sample (S_i) is used, how is this choice motivated? It'll be good to have some ablation study of the combined effect of using only one data sample in a mini-batch, and the full-explored masking. For example, if for the baseline model, we also only use one data sample and apply different masks, will there be improvement?

In experiments, mainly accuracy is shown, but since the major motivation to reduce gradient variance, why not show some comparison of gradient variance of MLM and MLM-FE?

In Sec2, you said "yet is outperformed by the proposed fully-explored masking (see Table 2).", do you mean Table 3?

In addition, I have several notation confusions:

Assumption1: What is the hamming distance m_1 and m_2, when m_i are random variables? Do you mean the expected hamming distance? Or we are assuming x is fixed? Please be more clear.

In sec3.1, you used S_t for minibatch, but in sec3.4, you use S_i for "a text sequence", which is confusing.

In sec4.1, the authors said "A batch size of 256 is employed, ", does that mean K=256 in Algorithm 1?

---

### Official Review · AnonReviewer1 · 2020-10-26
**Official Blind Review #1**

**Rating:** 4
**Confidence:** 5

**Review:**

Summary:

This work proposes a fully-explored masking strategy, which maximizes the Hamming distance between any of the two sampled masks on a fixed text sequence. The motivation is to reduce the undesirable large variance of MLM objective, based on the hypothesis that randomly sampled masks in MLM would lead to undesirably large gradient variance, which as a result typically hurts the training efficiency with stochastic gradient optimization algorithms.

---------------------------------------
Strength:

The hypothesis from the variance reduction is interesting. The method is sound. Theoretical discussion proves that the gradients derived from the new masking schema have a smaller variance and can lead to more efficient self-supervised training. Experiments on both continual pre-training and general pre-training from scratch show the effectiveness of the proposed method. Case studies show that the method can help improve training efficiency.

---------------------------------------

Concerns & Questions:

1. Regarding the proof, the notations in Section 3 are quite loose, especially for Section 3.1. It might be better to tighten them and put the proofs in the Appendix into the main body of this paper.

2. The authors claim the variance of the K-masks gradient can be reduced by decreasing the gradient co-variance between different masks, but this is just for a given sentence and the problem still exists considering different sentences.

3. The proof in A.2 is not persuasive, I cannot agree that equation 14 can be concluded through previous statements.

4. In terms of the experiments, there are only dev results reported on GLUE (Table 2). It is hard to infer the test gains, given the possibly significant hyperparameter optimization on the dev set. It seems that the authors have the infrastructure for computing single-model test-set results. So why not report the test results?

5. Why are the backbone models (RoBERTa and BERT, respectively) different in Table 1 and Table 2? I'm concerned whether these improvements will hold after optimizing BERT carefully like RoBERTa, or using more advanced backbone methods like ALBERT. Why not using a larger model (e.g., Roberta-large)?

6. There is no comparison with other public masking methods in Table 2, such as whole-word-masking, span masking, etc. Did you use dynamic masking as that was previous used in RoBERTa? How about the benefits compared with the proposed one?

7. Concerning the claim, “the proposed fully-explored masking strategies lead to pre-trained models with stronger generalization ability.”, it is not clear how the proposed method yields stronger generalization ability.

8. The ablations cannot serve the topic of this paper well. For the first ablation, it just gives out the performance of intermediate models on a single task. It might be better to give out the trend of training loss and validation loss. For the second ablation, why do all the larger splits lead to similar performance? Intuitively, doing more mask-then-predict procedures is better for learning language representations. The explanation at the end of Section 4 is not persuasive.

---------------------------------------
Minor issue: the citation format is not consistent, please check the usage of \citep{} and \citet{}.

---

### Official Review · AnonReviewer3 · 2020-10-28
**Improve Masked Language Model pre-training by reducing the variance using a new masking scheme.**

**Rating:** 6
**Confidence:** 4

**Review:**

This paper try to improve Masked Language Model pre-training by reducing the variance using a new masking scheme.

Pros:
1. It is very novel and interesting that the authors try to understand the training of MLM by analyzing the relation gradient variance and masking scheme.
2. They propose a new masking scheme that reduces the gradient variance.
3. This paper shows the gradient obtained by the new masking is not biased. (Lemma 2)
4. The performance improvement under continual pretraining and BERT pretraining looks good to me.
5. The convergence is faster than the traditional MLM scheme.


Cons/Questions:
1. One of my concerns is that in the usual MLM training, each sentence is observed very limited times (e.g., $\leq 3$). Under such a setting, the overlapping between different maskings won't have too many overlapping.
2. It would be better to validate the assumption by comparing the gradient variance between MLM training with and w/o the new masking scheme.
3. It is not clear to me how Theorem 1/Assumption 1 motivates the new masking scheme. What does it mean by " The covariance $Cov(g(m1),g(m2))$ is monotone decreasing in terms of the Hamming distance"? Do you mean "$Cov(g(m1),g(m2) | H(m1,m2)=\tau)$ is monotone decreasing in term of $\tau$"
4. In Figure 4 (left), it would be better to run the experiments multiple times and plot mean and variance.
5. If we compare the result of 6-split masking of ACL-ARC (Figure 4 right) and the baseline (DAPT) in Table 1, there is actually no performance gain. I would say selecting an appropriate splitting number is tricky. It would be helpful to present the 1-split performance (i.e., DAPT) for better understanding of the results.

Minor:
1. The text in Figure 2 can be larger.
2. Last paragraph on page 4: in terms od ==> of

---

### Official Review · AnonReviewer2 · 2020-11-03
**Experiments need improvements**

**Rating:** 5
**Confidence:** 3

**Review:**

The paper argues that randomly sampled masks in masked language model can lead to large gradient variance, and hence it proposes a new masking strategy called fully-explored masking with theoretical support to reduce the variance. Experimental results show that fully-explored masking outperforms random masking in most cases.

Strengths:
1. The proposed method seems theoretically sound.
2. The new masking strategy is simple and easy to implement.

Weaknesses:
1. The authors claim that the proposed masking strategy can reduce the gradient variance. However, there are no empirical results to *directly* support the claim.

2. The proposed masking strategy is empirically compared with random masking only. However, there are existing papers which try to improve random masking, e.g. "ELECTRA" and "Ernie: Enhanced representation through knowledge integration". The authors may need to compare with these papers.

---

### Official Review · AnonReviewer5 · 2020-11-08
**Interesting Observation with Small Empirical Gains**

**Rating:** 6
**Confidence:** 4

**Review:**


###  Summary
This paper theoretically shows that the gradient variance of the standard MLM (masked language modeling) task in BERT-style training depends on the covariance of the gradient covariance between different masks within the mini-batch. This paper then empirically shows that the covariance can be reduced by making the masks less overlapped. A modified version of MLM is proposed, which has been shown with a smaller gradient variance than the standard MLM. The experimental results show that the new masking strategy does lead to some gains on several benchmarks.


### Strong Points
- This paper rigorously shows that the gradient of MLM can be reduced by decreasing the covariance of the gradient covariance between different masks within the mini-batch. This is a very interesting observation and it is nice to see that this can be proved mathematically.
- This paper is well-written and very easy to follow.

- The experiment is also well done.

### Weak Points
- This paper is based on several important assumptions, none of which has been proved mathematically (although there are some empirical evidence)
   + Assumption #1: the model quality can be improved by reducing the variance of the gradient.
   + Assumption #2: the covariance of gradients can be reduced by increasing the Hamming distance between two different masks.
- The empirical gain of the proposed method seems to be quite small. Given that the variance of the reported result is so large, it is unclear to me whether those gains are real.

### Other Comments
- Honestly, I don’t quite see why the proposed method is related to continual pretraining, and why should it be included in the experiment section. Is it because you want to show that under the situation that “in-domain data are usually much more limited” the proposed method is still effective?

---

### Decision · Program_Chairs · 2021-01-07
**Final Decision**

**Decision:**

Reject

**Comment:**

This work proposes a fully-explored masking strategy, segmenting the input text, which maximizes the Hamming distance between any two sampled masks on a fixed text sequence. The hope is to reduce the large variance of MLM objective, based on the hypothesis that randomly sampled masks in MLM lead to undesirably large gradient variance, which typically hurts training efficiency with stochastic gradient optimization algorithms.

Pro

- A clear and interesting, novel leading theoretical idea. The paper has one good theme that it pursues.
- A mostly well-written paper
- Contains good theoretical discussion
- Experiments support the idea

Con

- The experiments could be better, especially they don't actually measure a reduction in gradient variance only accuracy
- The proofs are at best loose
- Alternative methods of reducing variance like using large mini batches are not considered
- The results might go away with use of stronger (larger) contextual LMs
- There isn't good comparison to other methods of masking like span masking and salient term masking
- Some of the things included seem quite haphazard (the ablations don't seem to the point, it's not really clear what this has to do with continual learning)

Overall, this paper feels to be in a premature state. The idea is interesting, but the idea and the paper needs to be developed more, with stronger results. I think it doesn't deserve to be accepted at this time.